# Gender Disparities in Vascular Access and One-Year Mortality among Incident Hemodialysis Patients: An Epidemiological Study in Lazio Region, Italy

**DOI:** 10.3390/jcm10215116

**Published:** 2021-10-30

**Authors:** Laura Angelici, Claudia Marino, Ilaria Umbro, Maurizio Bossola, Enrico Calandrini, Luigi Tazza, Nera Agabiti, Marina Davoli

**Affiliations:** 1Department of Epidemiology Regional Health Service Lazio, 00147 Rome, Italy; l.angelici@deplazio.it (L.A.); c.marino@deplazio.it (C.M.); e.calandrini@deplazio.it (E.C.); n.agabiti@deplazio.it (N.A.); m.davoli@deplazio.it (M.D.); 2Geramed Dialysis Center, Fiano Romano, 00065 Rome, Italy; 3Haemodialysis Unit, Department of Medical and Surgical Science, Policlinico Universitario Fondazione Agostino Gemelli, 00168 Rome, Italy; mauriziobossola@gmail.com; 4Catholic University, 00168 Rome, Italy; luigi.tazza@gmail.com; 5Ars Medica Clinic, 00191 Rome, Italy

**Keywords:** AVF, catheters, dialysis, gender, mortality, vascular access

## Abstract

(1) Background: Interest in gender disparities in epidemiology, clinical features, prognosis and health care in chronic kidney disease patients is increasing. Aims of the study were to evaluate the association between gender and vascular access (arteriovenous fistula (AVF) or central venous catheter (CVC)) used at the start of hemodialysis (HD) and to investigate the association between gender and 1-year mortality. (2) Methods: The study includes 9068 adult chronic HD patients (64.7% males) registered in the Lazio Regional Dialysis Register (January 2008–December 2018). Multivariable logistic regression models were used to investigate the associations between gender and type of vascular access (AVF vs. CVC) and between gender and 1-year mortality. Interactions between gender and socio-demographic and clinical variables were tested adding the interaction terms in the final model. (3) Results: Females were older, had lower educational level and lower rate of self-sufficiency compared to males. Overall, CVC was used in 51.2% of patients. Females were less likely to use AVF for HD initiation than males. 1354 out of 8215 (16.5%) individuals died at the end of the follow-up period. Interaction term between gender and vascular access was significant in the adjusted model. From stratified analyses by vascular access, OR female vs. male (AVF) = 0.65; 95% CI 0.48–0.87 and OR female vs. male (CVC) = 0.88; 95% CI 0.75–1.04 were found. (4) Conclusions: This prospective population-based cohort study in a large Italian Region showed that in females starting chronic HD AVF was less common respect to men. The better 1-year survival of females is more evident among those women with AVF. Reducing gender disparity in access to AVF represents a key point in the management of HD patients.

## 1. Introduction

Chronic kidney disease (CKD) is a global health burden and an important risk factor for cardiovascular disease. Indeed, in 2017, CKD affected 697.5 million of individuals worldwide [1].

According to the 2019 Annual Data Report by United States Renal Data System the overall prevalence of CKD (stages 1–5) increased from 13.8% in 2016 to 14.5% in 2017 [2]. This trend reflects not only the increasing number of incident cases but also the longer survival of patients with end-stage renal disease (ESRD) in renal replacement therapy. In Lazio Region (Central Italy, where Rome is located), the integrated use of the regional health information systems (HIS) databases allowed an estimation of CKD prevalence. In 2017, CKD affected 99.457 persons (55.8% males) in Lazio Region, with a crude prevalence of 1.76%, 2.06% among males and 1.50% among females [3], reflecting the European general sex distribution in hemodialysis (HD) despite the higher proportion of females in the general population [4,5].

Important gender differences were widely demonstrated in specific medical conditions, in particular in cardiology. These include differences in clinical features, risk factors, disease progression, and access to effective care between men and women [6,7]. Recently, knowledge that many aspects of CKD manifest differently between men and women is increasing. In particular, epidemiology, progression and management of CKD may be influenced by gender [5,8,9]. CKD is more common in women, probably due to their longer life expectancy compared to men [10] as well as due to the inaccurate GFR estimation that characterizes women [11]. However, women are less likely receiving HD [12] possibly due to the faster decline of kidney function experienced by men [13,14] in part related to psycho-socioeconomic factors and unhealthier lifestyles [15,16,17,18]. Furthermore, outcomes such as mortality are different between men and women. The female longer life expectancy in general population is demonstrated to be reduced in HD resulting in a cancelled or reduced survival advantage compared to males [8,9,19].

Longevity on HD is directly proportional to the quality of HD which in turn depends on the reliability and integrity of the vascular access [20]. Vascular access also showed sex-dependent distinctions with AVFs less prevalent in females compared to males [8,21,22]. Arteriovenous fistula (AVF) confers a survival benefit relative to central venous catheter (CVC) in the incident HD population [23,24,25]. Despite the Fistula First Initiative, which campaigned for greater use of AVF and set goals of at least 66% AVF use among prevalent HD patients [26] regardless of age or comorbidity status [27,28], ≥80% of ESRD patients in US initiate HD with a CVC [24,29,30]. 

In the Lazio Region, the Regional Dialysis and Transplantation Registry (RRDTL), active since 1994 (DGR n. 7940/1987) and established by Regional Law 9/2010, is a tool of great importance for epidemiological monitoring, regional health planning, evaluation of quality of care and promotion of scientific research activities [31]. The large majority of ESRD patients used HD as renal replacement modality and the RRDTL collects information on patients in chronic HD from the date of incidence, updating them annually. 

Using the data of the RRDTL, the aims of the present study are: (1) to evaluate the association between gender and type of vascular access used at the start of hemodialysis (AVF or CVC); (2) to investigate the association between gender and 1-year mortality. 

## 2. Materials and Methods

### 2.1. Source of the Data

The source of the data is the RRDTL which contains information on chronic HD patients treated in all public and private dialysis Centers accredited in Lazio Region (5,892,425 inhabitants ISTAT 2015). It collects information on socio-demographic and clinical conditions of patients, type of dialysis and pharmacological treatments. Dialysis Centers fill in the data of own chronic HD patients at the start of renal replacement therapy and update them at least once a year. Every change such as recovery of renal function, shift to another dialysis Center, kidney transplantation or death must be communicated. 

### 2.2. Study Population

The study includes ESRD patients aged 18 years and over who started chronic HD (incident patients) between January 2008 and December 2018. Exclusion criteria were, previous failed AVF and having AV graft.

### 2.3. Outcome and Follow-Up

The primary outcome of the study was the type of vascular access recorded at the start of hemodialysis: AVF or CVC. Gender, categorized as female and male, was considered as predictor. The secondary outcome is the 1-year mortality in incident patents between January 2008 and December 2017 to ensure a one-year follow-up for all patients. Data on mortality was obtained from the Regional Mortality Registry.

### 2.4. Exposure

Gender, categorized as male and female, was considered as the main exposure in this study.

### 2.5. Co-Variates

#### 2.5.1. Socio-Demographic and Clinical Variables

Based on the information reported in the RRDTL, the following socio-demographic variables were collected at the start of HD: age at incidence (19–49, 50–64, 65–74, 75–84, 85+) and educational qualification (no qualifications/elementary school/middle school, high school/degree and more). Furthermore, we considered both clinical variables such as body mass index (BMI < 18.0, 18.0 ≤ BMI < 25.0, 25.0 ≤ BMI < 30.0, BMI ≥ 30.0) and self-sufficiency (complete, little self-sufficient, not self-sufficient) [32,33] and comorbidities as follows: cardiovascular risk factor (hypertension, diabetes and obesity), heart disease (arrhythmia, coronary artery disease or heart failure), peripheral vascular diseases (disease of the circulatory system outside of brain and heart), cerebrovascular disease (ischemic stroke, cerebral aneurysms, vascular malformations, intracerebral hemorrhage), chronic obstructive pulmonary disease, cancer, lipid metabolism’s alteration (hypercholesterolemia >240 mg/dL or hypertriglyceridemia >400 mg/dL), neurological disease (dementia, cognitive behavioral disorders or psychiatric diseases). Cause of ESRD includes: unknown, diabetic nephropathy, renal vascular disease, glomerulonephritis, interstitial nephritis, tossic/pyelonephritis, other nephropathies (gout, irreversible cortical or tubular necrosis, nephrocalcinosis, hypercalcemic nephropathy, Balkan nephropathy, traumatic or surgical kidney loss, renal tuberculosis, kidney cancer), other systemic disease (amyloidosis, Wegener’s granulomatosis, systemic lupus erythematosus, myeloma, light chain nephropathy, Schonlein-Henoch purpura, scleroderma, Goodpasture syndrome, hemolytic uremic syndrome, including Moschowitz syndrome), cystic renal disease/familial nephropathy, renal malformation. The laboratory findings collected included: serum creatinine, hemoglobin (Hb), creatine phosphokinase (CPK), serum phosphate level, serum albumin.

#### 2.5.2. Care-Related Variables

Type of dialysis Center (public or private), pre-dialysis counselling (patients followed by renal units in the 6 months prior to the start of chronic HD).

#### 2.5.3. Laboratory Findings

Serum creatinine (mg/dL), Hb (g/dL), CPK (U/L), serum phosphate level (mg/dL), serum albumin (g/L).

### 2.6. Statistical Analysis 

#### 2.6.1. Association between Gender and Type of Vascular Access 

Socio-demographic, clinical and care-related variables are presented as percentage or mean ± std according to gender (male or female) and type of vascular access (AVF or CVC) for incident patients. Bivariate relationships between socio-demographic, clinical and care-related variables, on the one hand, and type of vascular access and gender on the other hand, were tested with the chi-square test or Fisher’s exact for categorical variables and by *t*-test or Wilcoxon rank sum test for continuous variables.

Univariable and multivariable logistic regression models were used to investigate the associations between gender and type of vascular access (AVF vs. CVC). Variables significantly associated with the outcome in the univariable analysis (*p*-value χ^2^ < 0.05) were introduced in the multivariable model with the stepwise procedure. Comparative risk estimates were expressed as odds ratios (OR) and 95% CI.

Interactions between gender and socio-demographic and clinical variables were tested adding the interaction terms in the final model. 

#### 2.6.2. Association between Gender and 1-Year Mortality

Socio-demographic, clinical, care-related and laboratory variables are presented as percentage or mean ± std, according to 1-year mortality for incident patients.

Bivariate relationships between socio-demographic, clinical, care-related and laboratory variables, on the one hand, and 1-year mortality on the other hand, were tested with the chi-square test or Fisher’s exact for categorical variables and by *t*-test or Wilcoxon rank sum test for continuous variables.

Univariable and multivariable logistic regression models were used to investigate the associations between gender and 1-year mortality. 

Variables significantly associated with the outcome in the univariable analysis (*p*-value χ^2^ < 0.05) were introduced in the multivariable model with the stepwise procedure. Comparative risk estimates were expressed as odds ratios (OR) and 95% CI.

As the unadjusted female-to-male mortality odds ratio might have been influenced by sex-dependent differences in the characteristics of the study population, it was explored the effect of adding to the univariable model different covariates significantly associated with mortality in the multivariable analysis or showing gender-specific differences. Moreover, since the sequence of adding adjustments may modify the results, we conducted a sensitivity analysis modifying the order of entry of the covariates.

Interactions between gender and socio-demographic and clinical variables were tested adding the interaction terms in the final model. 

Statistical significance was set at a 2-tailed *p*-value of 0.05. All data were analyzed using SAS version 9.4 (SAS Institute, Cary, NC, USA).

## 3. Results

### 3.1. Association between Gender and Type of Vascular Access 

In the cohort of 9068 incident chronic HD patients the majority was male (64.7%) with a mean age of 68.2 ± 14.3 years. 

The baseline characteristics of the entire population at the start of HD according to gender are shown in Table 1. Females were slightly older (age class 85+: 10.4 vs. 8.2%), with a lower educational level (73.5 vs. 61.8%) and a lower rate of complete self-sufficiency (40.3 vs. 50.4%) compared to males. Additionally, females had higher rates of obesity (17.5 vs. 12.0%), neurological disease (3.9 vs. 2.7), cardiovascular risk factors (79.5 vs. 77.0%), and higher rate of early referral to nephrologist (76.5 vs. 74.7) compared to males. Finally, females had lower rates of heart disease (30.6 vs. 39.4%), peripheral vascular disease (10.5 vs. 16.8%), cerebrovascular disease (12.0 vs. 14.8%), chronic obstructive pulmonary disease (11.6 vs. 15.6%), and lower rate of cancer (10.5 vs. 13.1) compared to males.

The baseline characteristics of the entire population at the start of HD according to type of vascular access are shown in Appendix A. Overall, CVC was the dominant HD access type, used in 51.2% of patients. 4422 out of 9068 patients started HD using the AVF and only 30% (*n* = 1344) were females. 

Patients who started HD using AVF were predominantly male (69.6 vs. 60.1%).

They were younger (age class 75–84: 29.4 vs. 33.1%; age class 85+: 6.2 vs. 11.7%), had higher educational level (37.2 vs. 31.1%), higher rate of complete self-sufficiency (58.0 vs. 36.1%) than patients starting HD with CVC.

Moreover, patients starting with AVF had higher rate of obesity (14.5 vs. 13.5%), cardiovascular risk factors (80.5 vs. 75.4%), higher rate of afference to private dialysis Centers (65.8 vs. 59.6) and higher rate of early referral to nephrologist (86.0 vs. 65.2) than patients starting with CVC.

In univariable logistic regression model females were less likely to use AVF for HD initiation over CVC than males (OR 0.66; 95% CI 0.60–0.72; *p <* 0.001) (Table 2). 

Covariates significantly associated at univariable level with type of vascular access, reported in Appendix A, were included in the multivariable model by stepwise-selection procedure.

After stepwise the covariates entered in the final model were: age class, BMI class, self-sufficiency, cardiovascular risk factors, heart disease, peripheral vascular disease, chronic obstructive pulmonary disease, cancer, type of dialysis center and early referral to nephrologist. 

In the final model after adjustment for significant covariates, females were less likely to use AVF for HD initiation than males (OR 0.67; 95% CI 0.61,0.73; *p <* 0.001) (Table 2). No significant interactions between gender and other independent variables were found.

### 3.2. Association between Gender and 1-Year Mortality

Table 3 shows characteristics of the study population by dead/alive status. 1354 out of 8215 individuals died at the end of the follow-up period. The mortality was 16.1% for males and 17.2% for females with an increasing trend for age classes (from 4.9% in 19–49 up to 31.6% in 85+). Furthermore, mortality was higher for subjects with a low level of education (18.2%), BMI < 18.0 kg/m^2^ (26.8%), not self-sufficient (35.3%), with an AVF as vascular access (25.7%) and without pre-dialysis counselling (19.1%). 

Results of univariable and adjusted OR for female to male mortality risk in incident HD patients are shown in Figure 1.

It was found that OR and 95% confidence interval of 1-year mortality of females versus males moved from 1.08 (95% CI 0.96–1.22, unadjusted baseline model) to 0.84 (95% CI 0.74–0.96) overall after adjusting for age at incidence, BMI, self-sufficiency, and cause of ESRD. This result indicates that females have a slightly lower mortality risk than males at similar age, BMI, self-sufficiency and cause of ESRD, and this result was statistically significant.

Subsequent adjustments for cancer, heart disease, peripheral vascular diseases, cardiovascular risk factors and laboratory findings (serum creatinine, serum albumin, Hb) altered the female-to-male mortality risk only marginally, with the exception of the adjustment for type of vascular access, which significantly decreased the female-to-male mortality risk (OR = 0.82; 95%CI 0.71–0.94).

The results of the sensitivity analysis, performed modifying the order of entry of the covariates, showed that the findings were consistent: adding adjustments for covariates showed a shift to the left (decreasing the OR for females versus males) with a significant decrease after the adding of type of vascular access.

There was no evidence of interaction between gender and other independent covariates except for vascular access (*p* = 0.034). From stratified analyses by vascular access OR female vs. male = 0.65; 95%CI 0.48–0.87 in those with AVF and OR female vs. male = 0.88; 95%CI 0.75–1.04 in those with CVC were found. 

Appendix A shows results from univariable and multivariable logistic regression on the association between gender (female vs. male) and type of vascular access (AVF vs. CVC) for HD initiation adjusted for socio-demographic, clinical, and care-related variables. 

## 4. Discussion

This large population-based cohort study shows a significant difference in the vascular access for HD initiation between men and women, in fact only 30% of females started using an AVF. Women experienced a significant better one-year survival in comparison to men and this advantage was greater among women who received AVF at HD initiation.

The evidence of disparity in vascular access between genders is not novel. In 2002, Reddan et al. demonstrated that the likelihood of starting HD with a CVC was higher in women and that, in the multivariate analysis, female gender was one of the independent variables predictive of CVC in the final logistic regression model [34]. Port et al., using data from an international study of dialysis patients, DOPPS, found that male gender was associated with an increased likelihood of AVF versus arterio-venous graft use among incident HD patients [35]. More recently, data from the United States Renal Data System (USRDS) revealed that the odds of AVF use at initiation of HD were significantly lower in women compared to men (OR = 0.69, 95% CI 0.67–0.71, *p* < 0.0001) with the gender gap in AVF use at initiation being highest in New York and the upper Midwest and smallest for Southern California and the Pacific Northwest and Alaska, suggesting variability related to cultural and organizing variables [22]. The study of Shah et al. showed similar findings, and women, relative to men, had 15% adjusted lower odds of AV access versus CVC [29]. In Europe, USA and Australia, but not in Japan, a higher use of CVC in women compared to men has been demonstrated [8]. In 2020 our group have already showed a likelihood to start HD with AVF significantly higher in males compared to females [36].

Overall, the reason of the differences in vascular access between men and women remains unclear. In general, AVF represents the best vascular access in HD and should be the first choice according to different guidelines [20,37,38], due to a better function with lower complications and mortality. It is well known that determining the optimal vascular access is based on a multitude of factors such as age, gender, comorbidities, life expectancy, patient anatomy, surgeon experience and organizing capacities of the single HD center [39,40]. The concerns about smaller vessels in women, leading to poorer AVF maturation and less long-term patency, may have prevented nephrologists and surgeons from considering AVF in female HD patients. Several studies have showed no differences in vasculature of females in HD by duplex ultrasound examination, with consequent adequacy of AVF placing [21,40]. Even if it seems that vein and arterial anatomy and caliber do not differ significantly by gender, some authors have hypothesized that the worst AVF outcome reported in women in some studies [21,41], but not in others [42], may persuade surgeons and nephrologists to prefer CVC. Finally, it is possible that aesthetic preferences may lead to the use of CVC in women. Some other possible explanations related to CVC preference in women might be related to cosmetic reasons as well as to low surgical training and experience [40]. It remains that the reduction of gender disparity in the access to AVF in patients initiating HD requires further investigation in the next future. 

The current study shows that females had a slightly lower mortality risk than males after adjustment for age, BMI, self-sufficiency and cause of ESRD, and this result was statistically significant. In addition, we observed a further reduction of mortality risk in females compared to males when considering only women who were offered AVF at HD initiation. Data on the effect of gender on mortality in incident patients are still conflicting. In some studies, gender was not a risk factor of one-year mortality [43,44,45,46]. Conversely, Shah et al [47], using the United States Renal Data System, reported that among 944.650 adult patients who initiated dialysis between 2005 and 2014, women experienced a survival advantage over men in 2005, but by 2014 it was reversed to survival advantage for men. However, combining all years, women had a 2% lower adjusted risk of dying at 1-year after dialysis initiation than men. A recent review by Carrero et al [9] showed a similar mortality for men and women despite a higher mortality among men at all levels of pre-dialysis CKD. Moreover, in our study, the use of AVF confers a lower 1-year mortality than the use of CVC. These data are in line with the recent literature. In 2017, Brown et al [48] demonstrated a 12-months mortality of 17% in the AVF group vs. 46% in the CVC group, adjusted for sex. Given the adjustment of the analysis for age, cause of ESRD, comorbidities, type of vascular access, and laboratory parameters, the lower one-year mortality observed, in the present study, in women initiating HD is difficult to explain and deserves further investigations.

Sex-dependent differences are well established in several clinical conditions, especially in cardiology [6,7]. We evaluated if sex-dependent differences in HD patients might exert some effects on variables such as patient care and mortality. The proportion of women in the general population is higher and CKD is more common in women than in men [9,49]. However, in HD population the proportion of women was only about 40%. Probably this difference is related to the faster progression of CKD in men [13] either to the shorter life expectancy that characterized females with ESRD before starting HD [10]. In particular, females are older and are characterized by less educational level associated with diabetes and cardiovascular events. In our study we confirmed the general trend of women to be older and to have low educational level together with low self-sufficiency. Furthermore, females are characterized by a slightly, but not significant, higher rate of early referral to nephrologist. Conversely, in literature, women are reported to be less aware of CKD, to initiate HD later than men and to die more before starting HD [10,50,51].

Furthermore, the higher survival rate characterizing females in the general population is markedly reduced in HD patients [19]. Men continued to have a slightly but significantly higher mortality risk than women at similar age and time on dialysis [8]. The survival advantage that women have over men in the general population was markedly diminished in HD where women’s survival was equal or slightly better than men’s survival. Furthermore, women start HD more frequently with CVC and are characterized by higher mortality due to access-related infections [52].

Strengths of this study are the large sample size and the integrated use of several Regional databases. These tools are subjected to continuous quality controlling that allow us to have accurate and standardized data. Generally, the epidemiological monitoring of CKD patients in Italy is relatively difficult because of the existing studies are related only to definite small areas and not to the entire country. Further potential confounders are probably the setting of the survey not allowing to collect accurate data, in particular a higher rate of response to survey was observed in the north versus the rest of Italy, in line with the higher economic and educational levels of this area [53]. In Lazio Region an accurate epidemiological monitoring was possible thanks to the integrated use of regional health information systems (HIS) databases and the Regional Mortality Registry. A limitation of the study is the lack of confounding parameters such as further laboratory parameters that may be useful for explaining the underlying pathophysiological mechanisms. Another limitation could be the existence of two populations under study. However, a comparative analysis of socio-demographic and clinical characteristics between the two populations did not highlight any differences and therefore the analytical approach was considered valid.

In conclusion, the present prospective population-based cohort study in a large Italian Region shows that in women initiating HD, with respect to men, AVF was less common, and that one-year mortality was significantly lower. The identification of the mechanisms underlying these results requires further studies in the future. Nevertheless, the need for sex-specific treatment strategies in ESRD care is suggested.

## Figures and Tables

**Figure 1 jcm-10-05116-f001:**
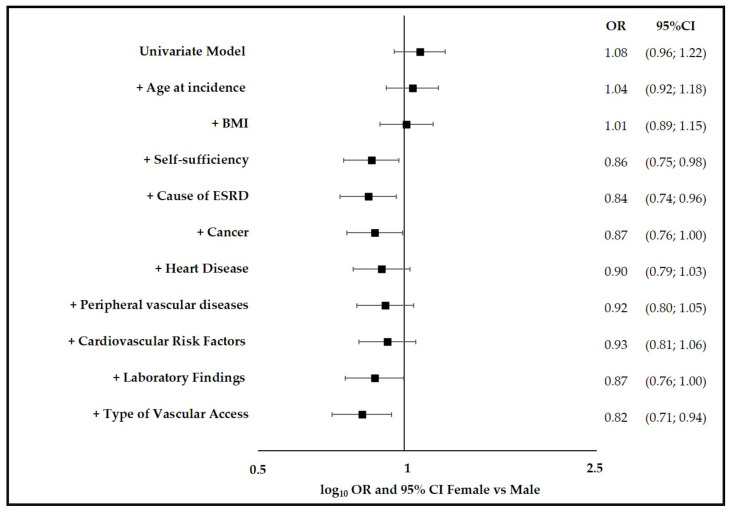
Univariable and adjusted OR for female to male mortality risk in incident hemodialysis patients from RRDTL Lazio Region (Italy) 2008–2017. RRDTL, the Regional Dialysis and Transplantation Registry, BMI, body mass index; ESRD, end-stage renal disease; OR, odds ratio; CI, confidence interval. Laboratory findings: serum creatinine, serum albumin, Hb. Cardiovascular risk factors: hypertension, obesity, diabetes.

**Table 1 jcm-10-05116-t001:** Distribution of socio-demographic, clinical, and care-related characteristics according to gender. Incident hemodialysis patients from RRDTL Lazio Region (Italy) 2008–2018.

		GENDER					
		**MALE**		**FEMALE**		**Total**	***p*-Value ***
		* **n** *	**%**	* **n** *	**%**	* **n** *	
**Total**		5868	100.0	3200	100.0	9068	
**Age class**	**(years)**						0.003
	**18–49**	714	12.2	380	11.9	1094	
	**50–64**	1260	21.5	674	21.1	1934	
	**65–74**	1595	27.2	793	24.8	2388	
	**75–84**	1816	31.0	1021	31.9	2837	
	**85+**	483	8.2	332	10.4	815	
**Age (years) mean ± std**		5868	68.1 ± 14.1	3200	68.5 ± 14.7	9068	0.124
**Education qualification**							<0.001
	**No qualifications/Elementary School/Middle School**	3627	61.8	2353	73.5	5980	
	**High School/Degree and more**	2241	38.2	>847	>26.5	>3088	
**Body Mass Index**							<0.001
	**Underweight (BMI < 18.0)**	208	3.5	283	8.8	491	
	**Normal weight (18.0 ≤ BMI < 25.0)**	3071	52.3	1561	48.8	4632	
	**Overweight (25.0 ≤ BMI < 30.0)**	1884	32.1	797	24.9	2681	
	**Obese (BMI ≥ 30.0)**	>705	>12.0	>559	>17.5	>1264	
**Self-sufficient**							<0.001
	**Complete**	2955	50.4	1289	40.3	4244	
	**Little self-sufficient**	1692	28.8	943	29.5	2635	
	**Not self-sufficient**	1221	20.8	968	30.3	2189	
**Comorbidities**							
	**Heart disease**	2313	39.4	980	30.6	3293	<0.001
	**Peripheral vascular diseases**	988	16.8	335	10.5	1323	<0.001
	**Cerebrovascular disease**	866	14.8	385	12.0	1251	<0.001
	**Chronic obstructive pulmonary disease**	915	15.6	372	11.6	1287	<0.001
	**Cancer**	770	13.1	336	10.5	1106	<0.001
	**Lipid metabolism’s alteration**	392	6.7	226	7.1	618	0.490
	**Neurological disease**	159	2.7	126	3.9	285	0.001
	**Cardiovascular Risk Factors (Hypertension, Obesity, Diabetes)**	4516	77.0	2545	79.5	7061	0.005
**Vascular access**							<0.001
	**AVF**	3078	52.5	1344	42.0	4422	
	**CVC**	>2790	>47.6	>1856	>58.0	>4646	
**Type of dialysis unit**							0.653
	**Public**	2201	37.5	1185	37.0	3386	
	**Private**	3667	62.5	2015	63.0	5682	
**Pre-dialysis counselling**							0.052
	**Yes**	4386	74.7	2448	76.5	6834	
	**No**	1482	25.3	750	23.4	2232	
**Laboratory findings, mean ± std**		** *n* **	**mean ± std**	** *n* **	**mean ± std**	**Total**	***p*-value**
	**Serum creatinine (mg/dL)**	5849	7.0 ± 2.3	3195	6.2 ± 2.0	9044	<0.001
	**Hb (g/dL)**	5865	10.3 ± 1.4	3198	10.1 ± 1.4	9063	<0.001
	**CPK (U/L)**	5868	8.7 ± 0.8	3200	8.7 ± 0.8	9068	0.009
	**Serum phosphate level (mg/dL)**	5822	5.0 ± 1.4	3169	5.0 ± 1.4	8991	0.023
	**Serum albumin (g/L)**	5840	3.6 ± 0.5	3185	3.5 ± 0.5	9025	<0.001

* Chi-square test or Fisher’s exact for categorical variable. *t*-test or Wilcoxon rank sum test for continuous variables. BMI, body mass index; AVF, arteriovenous fistula; CVC, central venous catheter; Hb, hemoglobin; CPK, creatine phosphokinase.

**Table 2 jcm-10-05116-t002:** Association between gender and type of vascular access (AVF vs. CVC) for hemodialysis initiation. Univariable and multivariable logistic regression.

		AVF	CVC								
	N	N (%)	N (%)	Univariable Model				Multivariable Model *			
Gender				Estimate	95% CI		*p*-Value	Estimate	95% CI		*p*-Value
Male	5868	3078 (52.5)	2790 (47.6)	_	_	_	_	_	_	_	_
Female	3200	1344 (42.0)	1856 (58.0)	0.656	0.602	0.716	<0.001	0.667	0.606	0.733	<0.001

* Multivariable analysis adjusted for age class, BMI class, self-sufficient, cardiovascular risk factors, heart disease, peripheral vascular disease, chronic obstructive pulmonary disease, cancer, type of dialysis center and early referral to nephrologist. AVF, arteriovenous fistula; CVC, central venous catheter; CI, confidence interval.

**Table 3 jcm-10-05116-t003:** Distribution of socio-demographic, clinical, and care-related characteristics according to dead/alive status at 1 year mortality. Incident hemodialysis patients from RRDTL Lazio Region (Italy) 2008–2017.

					1 YEAR DEATH				
		Dead			Alive			Total	*p*-Value *
		*n*	%col	%row	*n*	%col	%row	*n*	
**Total**		1354	100	16.5	6861	100	83.5	8215	
**Gender**									0.2205
	**Males**	855	63.2	16.1	4452	64.9	83.9	5307	
	**Females**	499	36.9	17.2	2409	35.1	82.8	2908	
**Age (years) mean ± std**		1354	74.8 ± 10.9		6861	67.0 ± 14.5		8215	
**Age class**	**(years)**								<0.001
	**19–49**	45	3.3	4.6	939	13.7	95.4	984	
	**50–64**	172	12.7	9.8	1576	23	90.2	1748	
	**65–74**	327	24.2	15.1	1835	26.8	84.9	2162	
	**75–84**	583	43.1	22.4	2019	29.4	77.6	2602	
	**85+**	227	16.8	31.6	492	7.2	68.4	719	
**Education qualification**									<0.001
	**No qualifications/Elementary School/Middle School**	992	73.3	18.2	4446	64.8	81.8	5438	
	**High School/Degree and more**	362	26.7	13	2415	35.2	87	2777	
**Body Mass Index**									<0.001
	**Underweight (BMI < 18.0)**	118	8.7	26.8	323	4.7	73.2	441	
	**Normal weight (18.0 ≤ BMI < 25.0)**	743	54.9	17.7	3449	50.3	82.3	4192	
	**Overweight (25.0 ≤ BMI < 30.0)**	357	26.4	14.7	2079	30.3	85.3	2436	
	**Obese (BMI ≥ 30.0)**	136	10	11.9	1010	14.7	88.1	1146	
**Self-sufficient**									<0.001
	**Complete**	247	18.2	6.6	3510	51.2	93.4	3757	
	**Little self-sufficient**	397	29.3	16.2	2049	29.9	83.8	2446	
	**Not self-sufficient**	710	52.4	35.3	1302	19	64.7	2012	
**Cause of ESRD**									<0.001
	**Unknown**	388	28.7	18.9	1668	24.3	81.1	2056	
	**Diabetic nephropathy**	318	23.5	17.1	1542	22.5	82.9	1860	
	**Renal vascular disease**	320	23.6	16.4	1629	23.7	83.6	1949	
	**Glomerulonephritis**	45	3.3	7	602	8.8	93	647	
	**Interstitial nephritis, toxic/pyelonephritis**	94	6.9	20.1	374	5.5	79.9	468	
	**Other nephropathies**	74	5.5	20.7	284	4.1	79.3	358	
	**Systemic disease**	83	6.1	28.5	208	3	71.5	291	
	**Cystic renal disease/familial nephropathy**	28	2.1	5.1	523	7.6	94.9	551	
	**Renal malformation**	4	0.3	11.4	31	0.5	88.6	35	
**Comorbidities**									
	**Heart disease**	683	50.4	22.7	2326	33.9	77.3	3009	<0.001
	**Peripheral vascular diseases**	242	17.9	25.9	692	10.1	74.1	934	<0.001
	**Cerebrovascular disease**	157	11.6	24.7	478	7	75.3	635	<0.001
	**Chronic obstructive pulmonary disease**	289	21.3	24.7	882	12.9	75.3	1171	<0.001
	**Cancer**	274	20.2	28.1	702	10.2	71.9	976	<0.001
	**Lipid metabolism’s alteration**	71	5.2	12.8	484	7.1	87.2	555	0.0153
	**Neurological disease**	79	5.8	30	184	2.7	70	263	<0.001
	**Cardiovascular Risk Factors (Hypertension, Obesity, Diabetes)**	944	69.7	14.9	5414	78.9	85.2	6358	<0.001
**Vascular access**									<0.001
	**FAV**	1043	77	25.1	3115	45.4	74.9	4158	
	**CVC**	311	23	7.7	3746	54.6	92.3	4057	
**Type of dialysis unit**									0.2191
	**Public hospital**	524	38.7	17.1	2534	36.9	82.9	3058	
	**Private clinic**	830	61.3	16.1	4327	63.1	83.9	5157	
**Pre-dialysis counselling**									0.0003
	**Yes**	975	72	15.7	5257	76.6	84.4	6232	
	**No**	379	28	19.1	1604	23.4	80.9	1983	
**Laboratory findings, mean ± std**		** *n* **	**mean ± std**		** *n* **	**mean ± std**		**Total**	***p*-value**
	**Serum creatinine (mg/dL)**	1349	6.0 ± 2.1		6845	6.9 ± 2.3		8194	<0.001
	**Hb (g/dL)**	1352	9.9 ± 1.3		6859	10.3 ± 1.4		8211	<0.001
	**CPK (U/L)**	1354	8.6 ± 0.8		6861	8.7 ± 0.8		8215	<0.001
	**Serum phosphate level (mg/dL)**	1338	4.7 ± 1.4		6809	5.0 ± 1.4		8147	<0.001
	**Serum albumin (g/L)**	1344	3.4 ± 0.5		6833	3.6 ± 0.5		8177	<0.001

* Chi-square test or Fisher’s exact for categorical variables. T-test or Wilcoxon rank sum test for continuous variables. BMI, body mass index; ESRD, end-stage renal disease; AVF, arteriovenous fistula; CVC, central venous catheter; Hb, hemoglobin; CPK, creatine phosphokinase.

## Data Availability

Data related to the findings reported in our manuscript are available to all interested researchers upon reasonable request and with the permission of the Regional Department, because of stringent legal restrictions regarding the privacy policy on personal information in Italy (national legislative decree on privacy policy no. 196/30 June 2003). For these reasons our dataset cannot be made available on public data deposition.

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
