# Peer review of "Gender Disparities in Vascular Access and One-Year Mortality among Incident Hemodialysis Patients: An Epidemiological Study in Lazio Region, Italy"

_jcm, 2021, doi:10.3390/jcm10215116_

Round 1

Reviewer 1 Report

This is an excellent study, using the large sample size of hemodialysis patients to show the gender disparities in vascular access and one-year mortality.

 The problem is not new but still not well understood, and no clear recommendations are helping to deal with it. Although the work is well written, it includes some concerns.

Major comments:

  1. In some groups of variables (especially age class, graduation, BMI, self-sufficiency...) tested for differences between classes, only one p-value is shown in tables. It is not clear if 1) all classes within the variable are compared with the one test; thus, one p-value is presented or 2) tests were performed for each pair of classes, and all p-values were equal.

In the more probable case 1), most results are presented incorrectly in the text because the presented p-value corresponds to the group (all tested pairs of classes) of the selected variable but not to the separate pair. Thus, the difference 2,2% ("age class 85+: 10.4 vs. 8.2%") could not have the same significance level as the difference 0,3% (age class 19-49: 12.2 vs. 11.9%). It means that the p-value tested for the group could not be considered for separate pair. In this case, the majority of results should be corrected, and a separate test for each pair of classes/variables has to be calculated.

In the less probable case 2), significance level should be expressed in each raw of compared pairs/variables, such as presented in comorbidities.   

Minor comments:

  1. In the abstract sentence: "...and OR Female vs. Male (CVC) = 0.88; 95% CI 0.93-1.04 were found" OR is incorrectly presented because its value is out of the CI range.
  2. The sentences in the abstract: "Females had a slightly and significantly lower mortality risk" and in the discussion: "The current study shows a slightly significant lower mortality in females" are not clear. What does it mean "slightly significant"? Slightly or significantly? or maybe "slightly, but significantly"? However, if p-value <0.1 but >0.05 this should be identified as a trend or significance level.

Reviewer 2 Report

The content is interesting. However, there are severe problems to be accepted. 

Method)

What is the definition of lipid metabolism’s alteration?

What is the difference between cerebrovascular disease and neurologic disease?

How did you get the information related to self-sufficiency? Via questionnaire? Please add the criteria dividing the categories of self-sufficiency and attach the reference file of the questionnaire.

Table 1)

Adding the information about the median age of both sexes might be helpful to understand the characteristics of enrolled patients. 

What is the definition of systemic disease as a cause of ESRD? Diabetes and hypertension are systemic diseases. 

What are the definitions of other nephropathies?

Why do the authors discriminate ‘cardiovascular risk factors’ summing hypertension, obesity, and diabetes? Respective categories are already introduced in the upper part of the table. 

Are there any patients using AV graft? 

It is better to add median (or average) age, Hb/HCT, Kt/V, comorbidity index, SBP/DBP, albumin, CRP between men and women.

Figure1)

Univariate OR did not show any sexual difference in 1-year mortality. Therefore, multivariate analysis is not needed. All categories (+ age at incidence, + BMI…) are needed to be definitely defined. For example, what are the laboratory findings?

Table 3)

Univariate OR did not show any sexual difference in 1-year mortality. Author should correct the mention of ‘Females had a slightly and significantly lower mortality risk than males..’ in abstract. 

This table simply analyzed the covariates (gender, age class, cause of ESRD…) using Table 1. I don’t know what we can gain from this table. Instead of this table, I wonder about the sexual difference in the cause of death. Is the mortality different in the patients with AVF or CVC? Author commented in abstract ‘better 1- year survival of females is more evident among those with AVF’. I cannot find the evidence.

All of the part should be corrected after accurate re-statistics. 

Font size and figure are different. It should be unified.

Reviewer 3 Report

This study is not original. Its design raises several questions

  • The study have 2 aims, but the authors use a different “n”: 9068 patients for evaluate the association between gender and vascular access; 8215 patients to investigate the association between gender and 1-year mortality
  • The variables are not defined

“Self-sufficiency”: how the authors defined complete, little and not self-sufficient

“heart disease”, “peripheral vascular disease” lacks definition

“lipid metabolism’ alteration” lacks definition

“cerebrovascular disease” and “neurological disease” what are the differences?

  • “Laboratory findings” When were collected?

Round 2

Reviewer 1 Report

The authors significantly improved their manuscript. 

I have no other concerns.

Reviewer 3 Report

The work was significantly improved and the limitations well described.